# ViewFusion: Learning Composable Diffusion Models for Novel View Synthesis

**Bernard Spiegl**[*]                                                                                   *bernard.spiegl@proton.me*
*Aalto University*

**Andrea Perin**                                                                                       *andrea.perin@aalto.fi*
*Aalto University*

**Stéphane Deny**                                                                                  *stephane.deny.pro@gmail.com*
*Aalto University*

**Alexander Ilin**[†]                                                                             *alexander.ilin@system2ai.com*
*System 2 AI*
*Aalto University*

**Reviewed on OpenReview:** *https://openreview.net/forum?id=amUisgrmte*

## Abstract

Deep learning is providing a wealth of new approaches to the problem of novel view synthesis, from Neural Radiance Field (NeRF) based approaches to end-to-end style architectures. Each approach offers specific strengths but also comes with limitations in their applicability. This work introduces *ViewFusion*, an end-to-end generative approach to novel view synthesis with unparalleled flexibility. *ViewFusion* consists in simultaneously applying a diffusion denoising step to any number of input views of a scene, then combining the noise gradients obtained for each view with an (inferred) pixel-weighting mask, ensuring that for each region of the target view only the most informative input views are taken into account. Our approach resolves several limitations of previous approaches by (1) being trainable and generalizing across multiple scenes and object classes, (2) adaptively taking in a variable number of pose-free views at both train and test time, (3) generating plausible views even in severely underdetermined conditions (thanks to its generative nature)—all while generating views of quality on par or even better than comparable methods. Limitations include not generating a 3D embedding of the scene, resulting in a relatively slow inference speed, and our method only being tested on the relatively small Neural 3D Mesh Renderer dataset. Code is available at `https://github.com/bronemos/view-fusion`.

## 1 Introduction

Novel view synthesis is a computer vision problem with a long research history. Traditionally, approaches that explicitly model the 3D space have been used such as voxels (Kim et al., 2013), point clouds (Agarwal et al., 2011) or meshes (Riegler & Koltun, 2020). With advancements of machine learning techniques and capabilities, various methods based on neural radiance fields (NeRFs) (Mildenhall et al., 2021) have emerged. These approaches aim to represent a 3D scene implicitly by using an MLP to parameterize it. Most recently, methods use an end-to-end, image-to-image approach where a collection of images of a scene is given to the model to produce a novel view of the scene (Sun et al., 2018; Dupont et al., 2020; Sajjadi et al., 2022b;a; 2023). Despite the extensive variety of methods, all the proposed approaches come with their drawbacks, such as (1) requiring expensive per-scene retraining and an abundance of input views, (2) an inability to

---

[*]Corresponding author. Work done at Aalto University, prior to appointment at CERN and is unrelated to the position.
[†]Work done at Aalto University.

Table 1: **Comparison of features with previous methods.** The features refer to the method's capability to: (1) operate in a setting where pose information about input views is not available, (2) generalize across multiple scenes and classes without the need to be retrained, (3) make use of variable input view count both at inference and training time, (4) produce a variety of plausible views when dealing with underdetermined, fully occluded target viewing directions.

| | LFN (Sitzmann et al., 2021) | PixelNeRF (Yu et al., 2021) | SRT (Sajjadi et al., 2022b) | ViT for NeRF (Lin et al., 2023) | **ViewFusion (Ours)** |
|---|---|---|---|---|---|
| 1) Pose-free | × | × | √ | √ | √ |
| 2) Generalization | √ | √ | √ | √ | √ |
| 3) Variable input | × | √ | × | × | √ |
| 4) Generative | × | × | × | × | √ |

operate without pose information about the input views or (3) an inability to adapt to a variable number of input views at test time. Therefore, the aim of this work is to introduce an intuitive end-to-end architecture for novel view synthesis which resolves the aforementioned drawbacks of previous work.

We propose *ViewFusion*, a novel approach that tackles the mentioned drawbacks all at once through a series of problem-specific design choices. Our method employs a diffusion probabilistic framework. We simultaneously apply a diffusion denoising step to any number of input views of a scene, then combine the noise gradients obtained for each view with a pixel-weighting mask, inferred specifically for every view, to ensure that for each region of the target view only the most informative input views are taken into account. Our method can be understood as a combination of multiple single-view denoising models for novel view synthesis, which, in addition to the noise predictions, produce weights used to aggregate the corresponding noise predictions of each of the single-view diffusion models during the denoising process. By training our method on a multitude of scenes and classes at once, we enable it to generalize without the need for retraining on every scene. Thanks to the stochastic nature of the diffusion process, the model is capable of performing well even in underdetermined settings (e.g. severe occlusion of objects or limited amount of input views) by providing a variety of plausible views. Our proposed solution does not require ordering nor any explicit pose information about the input views and, unlike the previous counterparts, once trained, the model is able to effectively handle inputs of arbitrary length. The ability to handle inputs of arbitrary length is thanks to the novel weighting mechanism that allows the model to weight views based on their informativeness and redundancy.

We evaluate our proposed approach on the Neural 3D Mesh Renderer (NMR) dataset (Kato et al., 2018; Chang et al., 2015) consisting of a wide variety of classes and input view poses. Through quantitative evaluation we show improved performance compared to relevant methods. We also qualitatively explore intermediate outputs of the model and confirm the soundness of our pixel-weighting mechanism to infer and adaptively adjust the importance of each of the input views: the inferred weighting scheme aligns with our human intuition that input views closer to the target view should be more informative than the further ones.

Summarized, the main contributions of this work are (also see Tables 1 and 2):

- a novel and intuitive approach to perform novel view synthesis, using a specifically tailored weighting mechanism paired with composable diffusion,

- a highly flexible solution thanks to the model's ability to process unordered and pose-free collections of images with variable length both at inference and training time, all while generalizing across a multitude of different classes,

- an inherent capability of the model to handle highly underdetermined (e.g. full occlusion) cases thanks to its generative capabilities,

- a competitive performance while providing significant flexibility improvements.

## 2 Related Work

Novel view synthesis is a long researched topic with solutions ranging from explicit modelling of the 3D space to more recent NeRFs and end-to-end approaches. However, these solutions often come with various drawbacks that our approach aims to address.

**Neural Radiance Fields (NERFs).** NeRFs (Mildenhall et al., 2021; Yu et al., 2021; Lin et al., 2023) aim to perform novel view synthesis by optimizing an underlying continuous volumetric scene function using a sparse set of input views. The volumetric scene function is represented by a neural network whose inputs are a spatial location in the form of position and viewing direction, and the output is color and volumetric density at a given point. Ultimately, once a NeRF model is trained, we can query all the possible spatial locations in order to obtain all possible views of the object. While NeRFs usually yield high quality results, they require a significant amount of training data for a single object and commonly do not generalize well across different objects, requiring object-dependent retraining. The need for per-scene retraining makes these models especially problematic when it comes to online applications where view synthesis has to be performed on objects of a previously unseen class. While there have been extensions that do not require per-scene retraining like Yu et al. (2021); Jiang et al. (2023), these approaches also rely on explicit and precise camera poses (Yu et al., 2021) which are not always available, or lack the generative capabilities to handle single-view cases or scenarios in which objects are occluded (Jiang et al., 2023).

**End-To-End Novel View Synthesis.** Using end-to-end based architectures is another commonly seen approach when it comes to novel view synthesis. For instance, Equivariant Neural Renderer (Dupont et al., 2020) aims to explicitly impose 3D structure on learned latent representations by ensuring that they transform like a real 3D scene. The encoded latents are then transformed before being passed to the decoder to produce the target view. Another example of an end-to-end approach that has shown promising performance for performing novel view synthesis is the Scene Representation Transformer (Sajjadi et al., 2022b) and its pose-free (Sajjadi et al., 2023) and object-centric (Sajjadi et al., 2022a) follow-ups. They focus on learning a latent representation of a scene by encoding a set of input images and passing it to a decoder to synthesise new views. A major drawback of the mentioned methods is their entirely deterministic nature, meaning that at inference time they do not have the ability to output a variety of plausible views when dealing with underdetermined scenarios, making them difficult to use in the settings such as severe occlusion or limited amount of input views. Sun et al. (2018) deal with this issue by employing a generative approach and self-learned confidence, but like the aforementioned NeRFs, they require explicit pose information to be provided for each of the input views.

**Diffusion Probabilistic Models.** Recently, models based on finding the reverse Markov chain transitions in order to maximize the likelihood of the training data, commonly known as diffusion probabilistic models (Sohl-Dickstein et al., 2015), have seen extensive use for solving various text-conditioned generative tasks (Ho et al., 2020; Rombach et al., 2022) including text-conditioned 3D synthesis (Poole et al., 2022; Li et al., 2023; Shi et al., 2023b). One of the main reasons for widespread use of diffusion models is their capability to produce high quality outputs when conditioned on textual descriptions. Diffusion models consist of a stochastic diffusion process and a deep neural network that parameterizes the denoising function used to perform the denoising procedure. Besides being used for generating images conditioned on text-based prompts, diffusion probabilistic models have seen applications in solving various other problems, including image-to-image novel view synthesis. These range from models that use pre-trained stable diffusion backbones such as Liu et al. (2023a) and its extensions (Shi et al., 2023a; Liu et al., 2023b; Chen et al., 2025), to models trained from scratch such as Müller et al. (2023), which operates directly on 3D radiance fields or Watson et al. (2022); Anciukevičius et al. (2023; 2024); Müller et al. (2024); Tang et al. (2024), in which a novel view diffusion based models are introduced to perform end-to-end novel view synthesis in image domain.

While these methods provide good qualitative results, they often come with various drawbacks. In particular, approaches based on using a pre-trained backbone rely on those backbones being freely and easily available. This is often not the case as authors of large models trained on extensive datasets refrain from making the weights and architectures publicly available, and instead provide access solely through an API, e.g. LLMs, large-scale diffusion models, etc. Furthermore, Müller et al. (2023) lacks the ability to generalize across multiple classes, Müller et al. (2024) is pose-conditional, Watson et al. (2022); Anciukevičius et al. (2023); Liu et al. (2023a); Shi et al. (2023a); Chen et al. (2025) cannot make use of additional views when they are available, Anciukevičius et al. (2024) lacks the ability to extrapolate at test time beyond number of input views available at training time and Tang et al. (2024) operates on a fixed grid with predetermined number of input views, resulting in reduced flexibility. Therefore, even though these approaches offer high quality synthesized views, they all come with specific drawbacks that we aim to address.

## 3 Method

Our approach (Figure 1) employs a composable diffusion probabilistic framework to generate novel views. The model receives an unordered, pose-free and arbitrarily long collection of input views $\{\mathbf{x}_i\}_{i=1}^N$ of a given scene along with the target viewing direction $\Delta\psi$. The model then predicts the scene as viewed from the target viewing direction $\Delta\psi$.

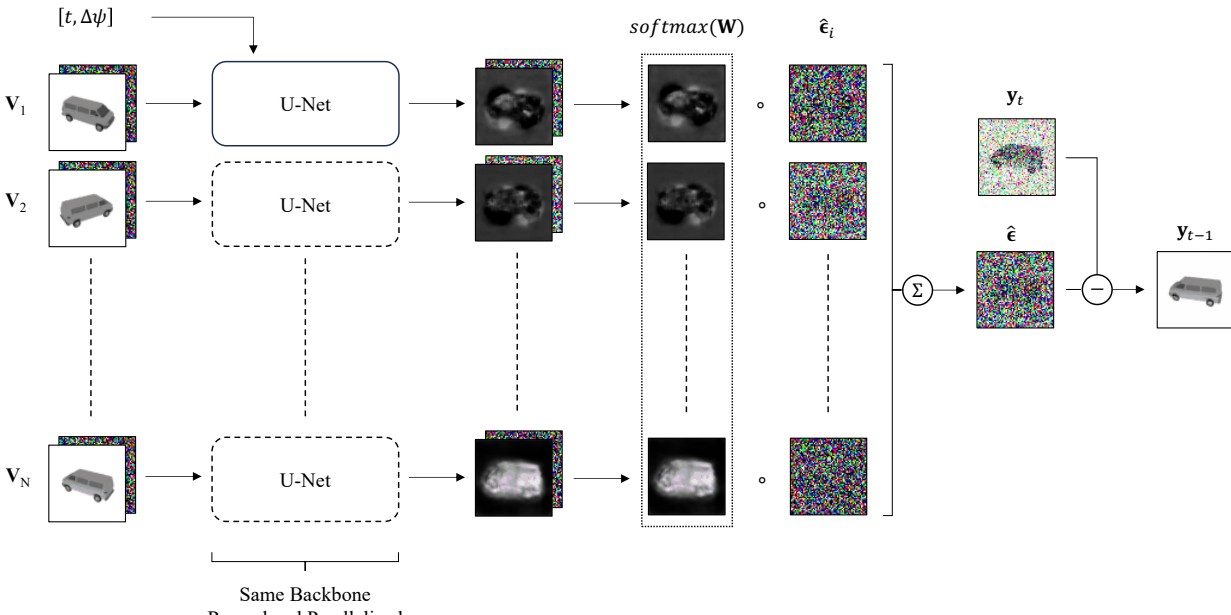

Figure 1: **Architecture Overview.** An arbitrary number of unordered and pose-free views coupled with the noise at timestep $t$ is denoised in parallel using a U-Net conditioned on timestep $t$ and target viewing angle $\Delta\psi$. The model then produces noise predictions and corresponding weights $\mathbf{w}_i$ for timestep $t$. A composed noise prediction, computed as a weighted sum of individual contributions, is then subtracted from the previous timestep prediction. Ultimately, after $T$ timesteps, a fully denoised target view is obtained.

### 3.1 Architecture

**Each input view is treated separately through identical streams.** We feed each input view separately to an identical copy of the denoising backbone of a diffusion model (see Appendix A for a technical description of diffusion). We use a U-Net as the denoising backbone, with the same architecture and hyperparameters as Saharia et al. (2022), unless specified otherwise below. Given a variable set of input views $\{\mathbf{x}_i \in \mathbb{R}^{3 \times \mathrm{H} \times \mathrm{W}}\}_{i=1}^N$ we concatenate the noisy image at timestep $t$, $\{\mathbf{y}_t \in \mathbb{R}^{3 \times \mathrm{H} \times \mathrm{W}}\}_{t=1}^T$, along the channel dimension to each

of the views, in order to produce a collection of U-Net inputs $\{\mathbf{V}_i \in \mathbb{R}^{6 \times \text{H} \times \text{W}}\}_{i=1}^N$, following Saharia et al. (2022). Additionally, in order to globally condition the model on the target pose information, positional encodings (Vaswani et al., 2017) of the target pose $\Delta\psi$ (single angle in radians) and timestep $t$ (integer) are concatenated and jointly embedded using a simple MLP consisting of two layers – one hidden layer with a *sigmoid* non-linearity followed by an output linear layer (dimensionality 128). The conditional embeddings produced by the MLP are injected into the U-Net through feature-wise affine transformations (Perez et al., 2018) in all downscaling and upscaling ResNet blocks. $\Delta\psi$ is the angular disparity between the target view and the canonical pose of the object, i.e. the front facing view of the object at 0°, as defined in the dataset (in Appendix D.1 we also explore relative canonical pose by defining $\Delta\psi$ as angular disparity between the target view and the first input view $\mathbf{V}_1$). We introduce the notation $\mathbf{c}_i^t := (\mathbf{x}_i, \mathbf{y}_t, \Delta\psi, t)$ to denote a tuple containing all the inputs to the U-Net.

**At each denoising step, all streams are composed through an inferred weighting strategy.** For a set of input views $\{\mathbf{x}_i \in \mathbb{R}^{3 \times \text{H} \times \text{W}}\}_{i=1}^N$, a set of corresponding noise weighting masks $\{\mathbf{w}_i \in \mathbb{R}^{3 \times \text{H} \times \text{W}}\}_{i=1}^N$ is produced at every timestep by the U-Net backbone. Indeed, the U-Net outputs pairs of noise predictions $\hat{\boldsymbol{\epsilon}}_i$ and weights $\mathbf{w}_i$, computed by two different heads at the final layer of the U-Net (obtained by changing the output layer channel count from 3 in the standard backbone to 6), each of the same dimensionality as the noise prediction (same dimension as the image). The weights tensor $\mathbf{W} \in \mathbb{R}^{(3 \times \text{N}) \times \text{H} \times \text{W}}$ is then normalized using *softmax* along the channel dimension. Next, these normalized weights are applied to the noise, and the weighted noise contributions are then summed, forming the final noise prediction $\mathbf{y}_{t-1}$. Intuitively, the weights reflect per-pixel informativeness of each of the input views for producing the target view. The described architecture is trained end-to-end using $L_2$ loss computed between the true noise and model's prediction. Algorithm 1 shows the training procedure pseudocode. $\alpha_t$ denotes the noise schedule and $\gamma_t = \prod_{t'}^t \alpha_{t'}$, further described in Appendix A.

---

**Algorithm 1** Composing View Contributions - Training

1: **repeat**
2:    $(\mathbf{x}, \Delta\psi, \mathbf{y}_0) \sim q(\mathbf{x}, \Delta\psi, \mathbf{y})$         {sample a data point}
3:    $t \sim \text{Uniform}(\{1, \ldots, T\})$         {sample a timestep}
4:    $\boldsymbol{\epsilon} \sim \mathcal{N}(\mathbf{0}, \mathbf{I})$         {sample noise}
5:    $\hat{\boldsymbol{\epsilon}}, \mathbf{W} \leftarrow f_\theta(\mathbf{x}, \sqrt{\gamma_t}\mathbf{y}_0 + \sqrt{1 - \gamma_t}\boldsymbol{\epsilon}, \Delta\psi, \gamma_t)$         {predict the noise and weights}
6:    $\mathbf{w}_i = softmax(\mathbf{W}, \text{dim}=0)_i$         {normalize the weights}
7:    Take gradient descent step on
8:       $\nabla_\theta \left\| \boldsymbol{\epsilon} - \sum_{i=1}^n \mathbf{w}_i \circ \hat{\boldsymbol{\epsilon}}_i \right\|^2$
9: **until** converged

---

At inference time (Algorithm 2), the denoising process is repeated for $T$ timesteps. Like in training, after each timestep, the noise predictions are weighted and summed together. Then, they are passed as conditioning for the next timestep prediction. Ultimately, after $T$ timesteps, the final target view is produced.

---

**Algorithm 2** Composing View Contributions - Inference

1: $\mathbf{y}_T \sim \mathcal{N}(\mathbf{0}, \mathbf{I})$         {sample starting noise}
2: **for** $t = T, \ldots, 1$ **do**
3:    $\mathbf{z} \sim \mathcal{N}(\mathbf{0}, \mathbf{I})$ if $t > 1$, else $\mathbf{z} = \mathbf{0}$         {sample noise}
4:    $\hat{\boldsymbol{\epsilon}}, \mathbf{W} \leftarrow f_\theta(\mathbf{x}, \mathbf{y}_t, \Delta\psi, \gamma_t)$         {predict the noise and weights}
5:    $\mathbf{w}_i = softmax(\mathbf{W}, \text{dim}=0)_i$         {normalize the weights}
6:    $\mathbf{y}_{t-1} = \frac{1}{\sqrt{\alpha_t}}\left(\mathbf{y}_t - \frac{1-\alpha_t}{\sqrt{1-\gamma_t}}\sum_{i=1}^n \mathbf{w}_i \circ \hat{\boldsymbol{\epsilon}}_i\right) + \sqrt{1-\alpha_t}\mathbf{z}$
7: **end for**
8: **return** $\mathbf{y}_0$

---

## 3.2 A Probabilistic Interpretation of ViewFusion

Here we provide a theoretical framework to justify the design choices of *ViewFusion* (see Appendix C for an extended version of this argument). We will do so by enforcing specific *desiderata* about the transition

probability of the reverse diffusion process, in the specific context of pose-free novel view synthesis. First, given a set of input views $\{\mathbf{x}_i\}_{i=1}^N$, the transition probability of the reverse diffusion process *should not depend* on the specific order in which these input views are fed to the model. Indeed, input views do not contain pose information, and thus cannot be ordered in any meaningful way. One way to enforce such permutation invariance is to write the transition probability of the reverse diffusion process as a sum of contributions for each of the input views, where each input view contributes separately through an identical energy function $E$:

$$p(\mathbf{y}_{t-1}|\mathbf{c}^t) \propto \sum_{i=1}^N \exp(-E(\mathbf{y}_{t-1}, \mathbf{c}_i^t)). \tag{1}$$

This functional form is permutation-invariant by construction (as also remarked by Zaheer et al. (2017)), hence it does not depend on the order of the $N$ input views. In addition, this functional form can be applied to any number of views, allowing our model to flexibly deal with an arbitrary number of views. This functional form can also be interpreted as a *mixture of experts*, where each individual view-conditioned stream acts as one expert in predicting the reverse diffusion process.

Next, we show how the softmax weighting scheme in our approach *directly derives* from the functional form described above, *without any further assumption*. The update step of a diffusion model is given by the *score* (Song et al., 2020b):

$$\nabla_{\mathbf{y}} \log p(\mathbf{y}_{t-1}|\mathbf{c}^t). \tag{2}$$

By replacing $p$ with the functional form proposed in equation 1, we show (derivation in Appendix C) that the score reduces to:

$$\nabla_{\mathbf{y}} \log p(\mathbf{y}_{t-1}|\mathbf{c}^t) = \sum_{i=1}^N \mathbf{w}_i \nabla_{\mathbf{y}}(-E(\mathbf{y}_{t-1}, \mathbf{c}_i^t)), \tag{3}$$

where $\mathbf{w}_i$ are the softmaxes over the energies $E_i$. This functional form for the update step is directly identifiable to the one we are using in our model, where (1) the weighting masks produced by our model correspond to the respective $\mathbf{w}_i$, and (2) the predicted noise for each view-dependent stream corresponds to its respective view-dependent score $\nabla_{\mathbf{y}}(-E(\mathbf{y}_{t-1}, \mathbf{c}_i^t))$. With this equivalence, we establish that the parallel stream architecture of *ViewFusion*, combined with its softmax aggregation scheme, derives directly from reasonable assumptions on the functional form of the transition probability of the reverse diffusion process, namely that it should be view-permutation-invariant, and more specifically that it should combine view-conditioned-predictions through a *mixture of experts* model.

## 4 Experimental Results

We evaluate our method on a relatively small, but diverse dataset, NMR, consisting of a variety of scenes and spanning multiple classes. We show that our model is capable of handling a wide variety of settings, while offering performances near or above the current comparable methods.

### 4.1 Dataset

**Neural 3D Mesh Renderer Dataset (NMR).** NMR has been used extensively in the previous works (Lin et al., 2023; Sajjadi et al., 2022b; Yu et al., 2021; Sitzmann et al., 2021) and serves as a good benchmark while keeping the computational footprint relatively low. The dataset is based on 3D renderings provided in Kato et al. (2018) and consists of 13 classes (sofa, airplane, lamp, telephone, vessel, loudspeaker, chair, cabinet, table, display, car, bench, rifle) from ShapeNetCore (Chang et al., 2015) that were rendered from 24 azimuth angles (rotated around the vertical axis) at a fixed elevation angle using the same camera and lighting conditions. The resolution of each image is $64 \times 64$. In total, there are 44 k different objects, split across training, validation and testing sets as follows: 31 k, 4 k, 9 k. There are no overlaps in individual objects between the sets.

## 4.2 Evaluation Procedure

In order to ensure good generalization performance both across a multitude of classes as well as variable input view count, we train our model by randomly picking the number of views that the model receives as conditioning, while simultaneously training across all available classes. At evaluation time we test our model both in fixed as well as variable input view count settings. Implementation details as well as training configurations are available in Appendix B.1, and the full evaluation procedure is described in Appendix B.2.

We limit our model to receive anywhere between one and six input views at random during the training, since providing an abundance of views can make the problem overly easy and completely determined while also requiring significant increase in computing power to process all of the input views. The NMR dataset provides 24 views for each object, viewed from the same elevation and rotated around the vertical axis. First, the number of views used for conditioning is uniformly sampled. Following this, we randomly select a subset of input views to be used for conditioning. We do not employ a specific sampling strategy, i.e. closer views to the target view are equally as likely as the further away ones.

However, even though we limit the training and inference to only up to six views, we also show that our approach is capable of generalizing to an arbitrary, previously unseen view counts.

NB **All the evaluations were performed on the same model, which was limited to receiving between one and six views at random during training time.**

## 4.3 Quantitative Results

We evaluate our model (Table 2) both in single-view and variable-view settings on commonly used metrics for novel view synthesis - PSNR, SSIM (Wang et al., 2004) and LPIPS (Zhang et al., 2018) on which we compare it to the most recent methods for novel view synthesis.

In order to ensure equivalence, we constrain our model to receiving only a single input view. This is effectively equivalent to turning off our learned weighting mechanism since all the weights scale to one after applying *softmax*. Even in this constrained scenario, our approach is able to reach competitive performance in LPIPS when compared to the previous approaches.

Table 2: **Quantitative Results.** Comparison of evaluated metrics against comparable models for NMR. NB To ensure equivalence of the results when comparing against SRT and ViT for NeRF, our model is restricted to receiving only a single input view at evaluation time. See Appendix B.2 for full evaluation procedure.

|  | ↑PSNR | ↑SSIM | ↓LPIPS |
|---|---|---|---|
| LFN (Sitzmann et al., 2021) | 24.95 | 0.870 | - |
| PixelNeRF (Yu et al., 2021) | 26.80 | 0.910 | 0.108 |
| SRT (Sajjadi et al., 2022b) | 27.87 | 0.912 | 0.066 |
| ViT for NeRF (Lin et al., 2023) | 28.76 | 0.933 | 0.065 |
| **ViewFusion (Ours)** - single view | 26.0 | 0.883 | **0.053** |
| **ViewFusion (Ours)** - up to six | **29.03** | 0.925 | **0.033** |

Additionally, we use the same model (without retraining) to compute the metrics for the setting where it receives anywhere between one and six views as input conditioning at random. By doing so, we reach an even better result in LPIPS, outperforming the current best approaches (to our knowledge), and on par with them when it comes to PSNR and SSIM. This goes to show that our model is capable of effectively utilizing the availability of additional views. Better LPIPS results can be attributed to the fact that our model does not produce blur in areas of uncertainty (e.g. occluded areas) thanks to its generative capabilities, unlike prior methods, which often suffered from blurry results when producing areas of uncertainty (Sitzmann et al., 2021; Yu et al., 2021; Sajjadi et al., 2022b). LPIPS also more accurately reflects human perception than the other two metrics.

We also explore how the performance scales as the input view count incrementally increases. We observe that the model's performance consistently increases with the view count even beyond the view count of six present

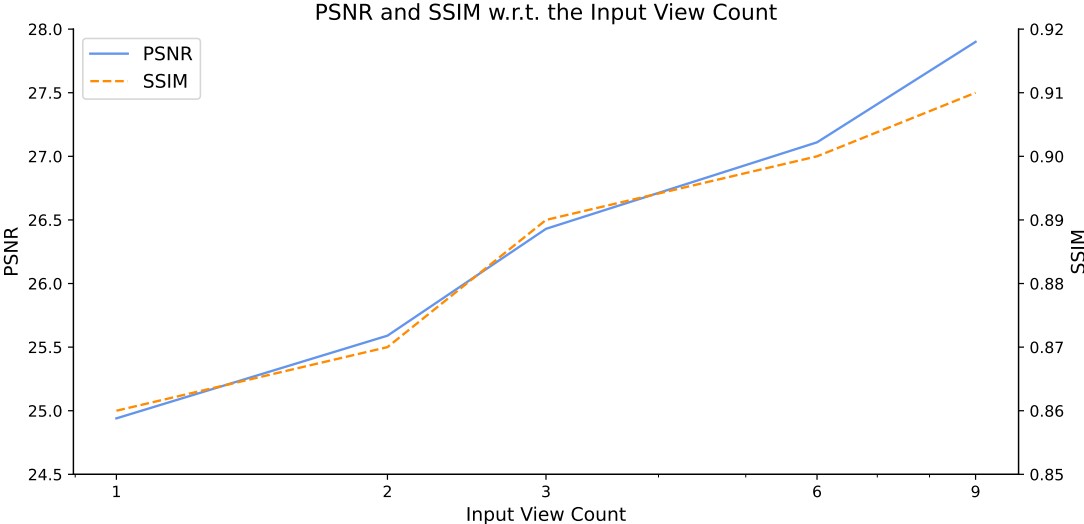

Figure 2: **Performance w.r.t. the Input View Count.** The model's performance consistently increases with the view count, even beyond the view count of six present at training time.

at training time, as shown in Figure 2. In this setting we use a portion of the test set for computational reasons.

### 4.4 Qualitative Results

In order to underline the flexibility of our approach, we subject it to a variety of different scenarios (Table 1).

**Variable Input Length.** One of the main advantages of our approach is the ability to effectively make use of a variable input view count, both during inference and training. Figure 3 shows that we are consistently able to produce high quality samples regardless of the input view count. Additionally, thanks to the model being class agnostic, we use a single model to produce the results across all of the classes.

**Adaptive Weight Shifting.** By altering the target viewing direction, we show that the model shifts the weighting adaptively according to the informativeness of the input views that are provided (Figure 4). Our weighting strategy allows the model to perform well even in scenarios where less informative views are given, that could normally act as a distraction. In Figure 4, we note that the views closest to the target view are selected by the weights, in accordance with our intuition that these views are most informative about the target view.

**Severe Occlusion.** Due to the generative nature of our approach, the model is capable of producing several plausible views when required to generate target views viewed from a direction that is occluded in the input views. Figure 5 shows outputs of the model conditioned on the same input view and diverse starting noise, reflecting scenarios where parts of the object present in the target view are heavily occluded in the input view.

**Autoregressive 3D Consistency.** Despite not imposing any explicit 3D consistency constraints, in Figure 6 we show that our approach is capable of maintaining 3D consistency through autoregressive generation even when provided solely with a single input view. We start by priming the model with a single input view, and incrementally rotate the target viewing direction to produce novel views. During the autoregressive generation, we fully utilize the flexible context length by adding each consecutively generated view to the conditioning for producing the next view. By doing so, we ensure that the model is 3D consistent with itself.

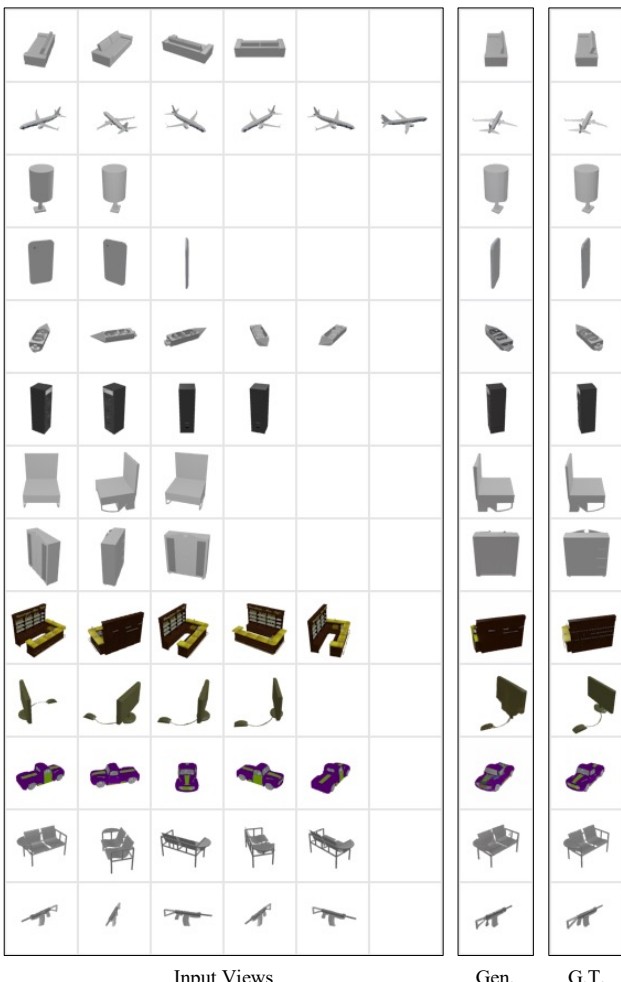

Input Views           Gen.   G.T.

Figure 3: **Variable Input Length.** Our approach is capable of generalizing across different classes without retraining. The padded empty views are included for visualization purposes only and are not present during training or inference, as the model operates with an unordered, pose-free view collection of variable length within the batches.

Additionally, unlike the previous approaches which often suffer from error accumulations by the time they reach the last frame, our model only elicits significant dissimilarities to the target when producing fully occluded parts (i.e. half way through the generative loop), which can not possibly be inferred from the initial input. Such behavior is caused by the the adaptive weighting putting more weight back on the initial view past the midpoint of generation and ensuring that the produced samples are consistent start to end.

**Generalization to Unseen View Counts.** The novel learnable weighting mechanism for composing the view contributions scales seamlessly to an arbitrary number of views that is even larger than the maximum view count presented during the training. In Figure 7, we show that our model trained on up to six views sensibly extrapolates even when presented with collections consisting of upwards of 20 views. Even though the problem is clearly determined with that many views, it is important to note that the model is able to sensibly utilize each of the views' contributions to produce the target view, all while maintaining its performance.

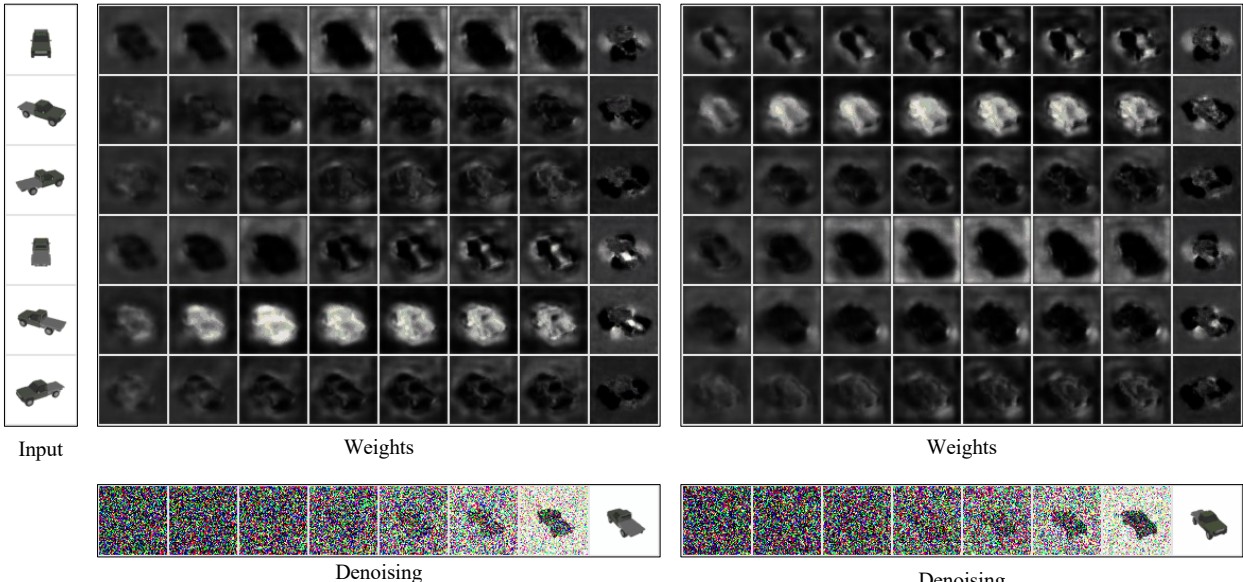

Input          Weights          Weights

Denoising          Denoising

Figure 4: **Adaptive Weight Shifting.** The model shifts its weighting adaptively based on the most informative input view w.r.t. the desired target output. Six views of the truck rotated by a fixed angular increment are passed in, depending on the target view the model puts most emphasis on the closest views. Additionally, the model picks up on the details from different views, e.g. the cargo bed and the back window of the truck are picked up from the fully rear facing input view in the example on the left or, in the example on the right, the front hull area gets picked up from the front facing input view.

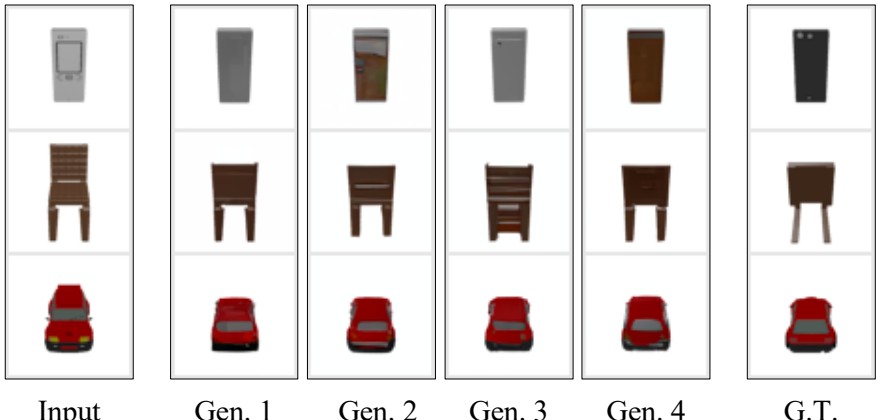

Input      Gen. 1      Gen. 2      Gen. 3      Gen. 4      G.T.

Figure 5: **Severe Occlusion.** Our approach is able to handle severely underdetermined settings by generating a variety of plausible view in cases where the target viewing direction is fully occluded. We prompt the model with the same front views several times to generate a variety of plausible rear views.

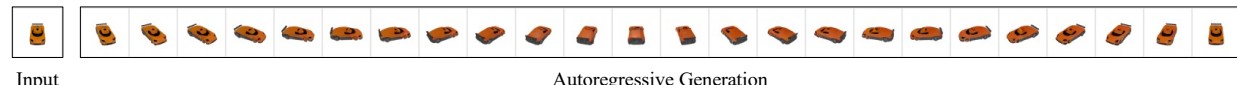

Input                                    Autoregressive Generation

Figure 6: **Autoregressive 3D Consistency.** Our approach is capable of maintaining 3D consistency through autoregressive generation even when provided solely with a single input view. We start by priming the model with a single input view, and incrementally rotate the target viewing direction to produce novel views. During the autoregressive generation, each consecutively generated view is added to the flexible conditioning for producing the next view.

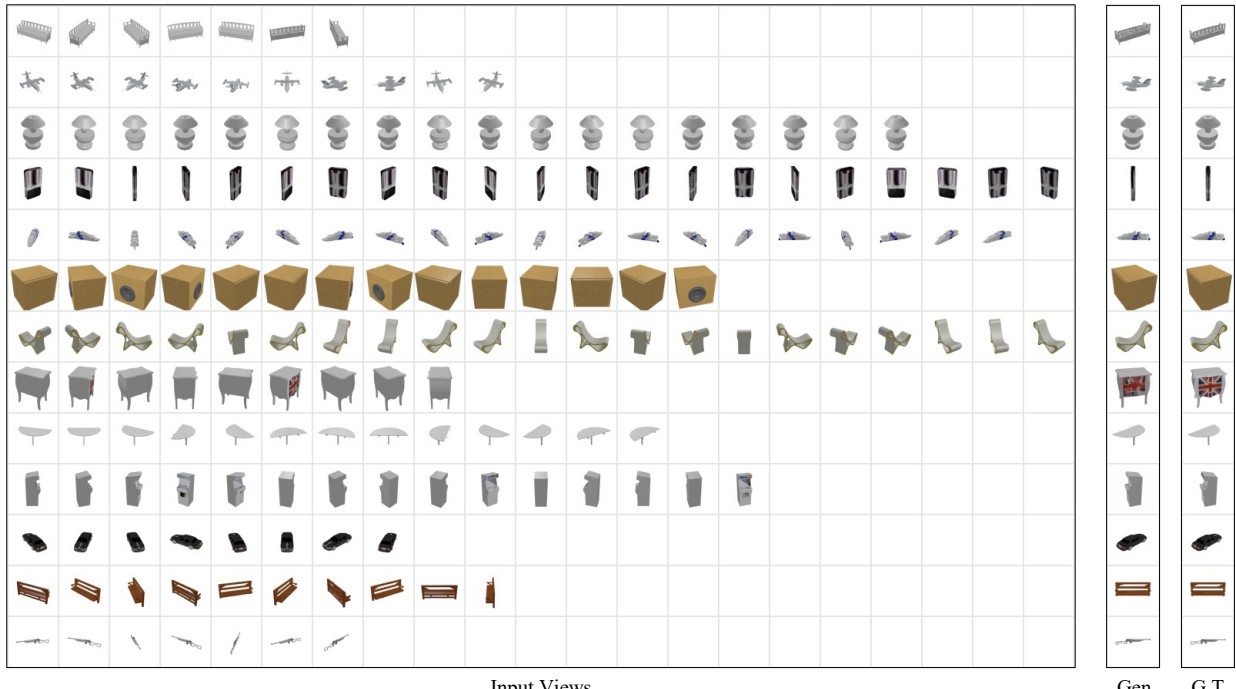

Input Views                                    Gen.    G.T.

Figure 7: **Generalization to Unseen View Counts.** In addition to taking an arbitrary number of input views, the model performs exceptionally well even when presented with significantly more input views at inference compared to the maximum of six at training time. Here we condition our model trained on up to six views on a significantly larger number of views.

## 5 Applications

This section explores potential application scenarios of ViewFusion based on findings of this work.

**Generating 3D Representations.**   As shown previously in Section 4.4, our model is capable of autoregressively synthesizing views of an object from all sides given only a single input view. Such capability could be useful for a wide variety of VR or AR applications, as well as when trying to build a 3D representation of an object based on one or few images (e.g. for creating video game assets). This would, however, require pairing with an approach that goes from image to 3D representation domain, as our method currently operates only in the image domain.

**Occlusion Prediction.**   Generative capabilities of our model could be leveraged to generate multiple plausible views of an occluded object, making the approach particularly useful in situations where any kind of plausible view is needed, even if it might not be ground truth, or in scenarios where the absolute correctness of the produced views is not of significant importance.

**Dataset Augmentation.**   Recent findings of Abbas & Deny (2023) have shown that commonly used deep networks for image classification fail to classify objects correctly when they are presented in unusual poses (e.g. flipped upside down or rotated in the 3D space). Therefore, if successfully scaled up, our method could be leveraged to produce more realistic results which could then be used to augment the already existing datasets used to train large classifiers, with the goal of improving their performance in a wide variety of edge-cases.

## 6 Limitations and Future Work

Our current approach does not explicitly incorporate 3D semantics of a scene, potentially posing a problem in situations where a quick, on-the-fly adaptation to a completely new, out-of-distribution scene is needed. When it comes to target pose conditioning, Appendix D.1 shows a promising step towards using relative canonical pose defined as the first input view, rather than defining it as the first, front facing view of the object at 0°. Another limitation is the trade-off between the generative power in underdetermined settings and the inference times for producing novel views, which scale linearly with the view count. Such scaling can make the model particularly slow when presented with a significant amount of input views or if the image resolution is significantly increased, especially when operating in the autoregressive mode. A potential way of fixing this would be to incorporate a well-established approach of performing diffusion in a latent space (Rombach et al., 2022) instead of image space. To further increase the inference speed, DDIM (Song et al., 2020a) sampling strategy could be employed instead of DDPM. Furthermore, our method does not explicitly impose constraints to ensure 3D consistency due to the stochastic nature of diffusion modelling. However, this can be partly alleviated by using autoregressive generation where each new view is also conditioned on previously generated views. Lastly, we test our model on a fairly limited and small NMR dataset. Therefore, to unleash the full potential of our generative approach and to enable it to operate in real-world scenarios, training on a more realistic and larger dataset, such as Objaverse (Deitke et al., 2023), CO3D (Reizenstein et al., 2021) or MVImageNet (Yu et al., 2023) would be a good future direction. Relatedly, our method can easily be implemented on top of already existing diffusion-based methods for novel view synthesis such as Liu et al. (2023a).

## 7 Conclusion

This work introduces *ViewFusion*, a flexible, pose-free generative approach for performing novel view synthesis using composable diffusion models. We propose a novel weighting scheme for composing diffusion models ensuring that only the most informative input views are taken into account for prediction of the target view, and enabling *ViewFusion* to adaptively handle an arbitrarily long and unordered collection of input views without the need to retrain. Additionally, the generative nature of *ViewFusion* enables it to generate plausible views even in severely underdetermined conditions. We believe that our approach serves as a

valuable contribution when it comes to novel view synthesis, with a potential of being applied to other problems as well.

**Broader Impact Statement**

We provide a brief discussion on potential impacts and considerations that our approach entails.

**Applications.**  As already outlined in Section 5, we believe that our generative model can be used for a wide variety of purposes, such as various VR and AR applications, as well as when building 3D models of objects given one or few images.

**Fake Content.**  This paper proposes a generative method, which could potentially be used to produce images containing fake or misleading content. We believe that given the relatively small scale of our current model, it poses no immediate threat. However, in a scenario where the approach would be expanded to a significantly larger or a more realistic datasets, the risks could be mitigated by using the model in a controlled environment.

**Energy Consumption.**  We propose a method that requires a training procedure which can be computationally expensive, particularly if applied to a larger dataset than NMR. Additionally, our current sampling procedure can require significant amount of computing power, especially if a large collection of views is used as conditioning. Despite the potential computational footprint of our solution, our method offers unparalleled flexibility without the need to retrain. Therefore, we believe that it still remains reasonably efficient, as training is by far the most expensive part of the procedure.

**Acknowledgments**

This work has been supported by a Helsinki Institute for Information Technology (HIIT) grant to B. Spiegl, and an Academy of Finland (AoF) grant to S. Deny under the AoF Project: 3357590. We would like to extend our gratitude to Aalto Science-IT project for providing the computational resources and technical support.

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

# A  Diffusion Probabilistic Models

In this appendix we briefly introduce the technicalities of vanilla diffusion probabilistic models (Ho et al., 2020) based on Saharia et al. (2022) and using the same notations as in the rest of the paper, but without the modifications introduced in Section 3.

Diffusion probabilistic models consist of a forward diffusion process during training and a corresponding reverse denoising process at the inference time. The forward process consists in gradually adding Gaussian noise to the image over $T$ timesteps, which can be described as following:

$$p(\mathbf{y}_t|\mathbf{y}_{t-1}) = \mathcal{N}(\mathbf{y}_t; \sqrt{\alpha_t}\mathbf{y}_{t-1}, (1-\alpha_t)I) \tag{4}$$

$$p(\mathbf{y}_{1:T}|\mathbf{y}_0) = \prod_{t=1}^{T} q(\mathbf{y}_t|\mathbf{y}_{t-1}) \tag{5}$$

where $\alpha_t$ is the noise schedule hyper-parameter. The noise is added up to the point where it is impossible to tell $\mathbf{y}_t$ from the Gaussian noise. The forward process at each step can also be marginalized as:

$$p(\mathbf{y}_t|\mathbf{y}_0) = \mathcal{N}(\mathbf{y}_t; \sqrt{\gamma_t}\mathbf{y}_0, (1-\gamma_t)I) \tag{6}$$

where $\gamma_t = \prod_{t'}^{t} \alpha_{t'}$. By applying Gaussian parametrization of the forward process, we obtain a closed form expression for the posterior distribution of $\mathbf{y}_{t-1}$ given $(\mathbf{y}_0, \mathbf{y}_t)$:

$$p(\mathbf{y}_{t-1}|\mathbf{y}_0, \mathbf{y}_t) = \mathcal{N}(\mathbf{y}_{t-1}|\boldsymbol{\mu}, \sigma^2\mathbf{I}) \tag{7}$$

in which $\boldsymbol{\mu} = \frac{\sqrt{\gamma_{t-1}}(1-\alpha_t)}{1-\gamma_t}\mathbf{y}_0 + \frac{\sqrt{\alpha_t}(1-\gamma_{t-1})}{1-\gamma_t}\mathbf{y}_t$ and $\sigma^2 = \frac{(1-\gamma_{t-1})(1-\alpha_t)}{1-\gamma_t}$. Having defined all the necessary aspects to perform diffusion, we can separate it into the training and inference procedures. During the training, the model acts as a denoising function and learns to invert the forward process. That means that given the noisy image $\tilde{\mathbf{y}}$, defined as:

$$\tilde{\mathbf{y}} = \sqrt{\gamma}\mathbf{y}_0 + \sqrt{1-\gamma}\boldsymbol{\epsilon}, \ \boldsymbol{\epsilon} \sim \mathcal{N}(0, \mathbf{I}) \tag{8}$$

the goal is to recover the original, target image $\mathbf{y}_0$. Therefore, the deep neural network model is parameterized as $f_\theta(\mathbf{x}, \tilde{\mathbf{y}}, \gamma)$, meaning it is conditioned on the input $\mathbf{x}$, a noisy image $\tilde{\mathbf{y}}$, and the noise level at a given time-step $\gamma$. The objective of the training procedure is to maximize a weighted variational-lower bound on the likelihood (Ho et al., 2020) and is given by:

$$\mathbb{E}_{(\mathbf{x},\mathbf{y})}\mathbb{E}_{\boldsymbol{\epsilon},\gamma}\left\| f_\theta(\mathbf{x}, \underbrace{\sqrt{\gamma}\mathbf{y}_0 + \sqrt{1-\gamma}\boldsymbol{\epsilon}}_{\tilde{\mathbf{y}}}, \gamma) - \boldsymbol{\epsilon} \right\|_p^p. \tag{9}$$

Pseudocode for the training procedure is given in Algorithm 3.

---

**Algorithm 3** Training - Forward (Noising) Process

1: **repeat**
2:     $(\mathbf{x}, \mathbf{y}_0) \sim q(\mathbf{x}, \mathbf{y})$
3:     $t \sim \text{Uniform}(\{1, \ldots, T\})$
4:     $\boldsymbol{\epsilon} \sim \mathcal{N}(\mathbf{0}, \mathbf{I})$
5:     Take gradient descent step on
6:         $\nabla_\theta \left\| f_\theta(\mathbf{x}, \sqrt{\alpha_t}\mathbf{y}_0 + \sqrt{1-\gamma_t}\boldsymbol{\epsilon}, t) - \boldsymbol{\epsilon} \right\|^2$
7: **until** converged

---

In order to perform inference, we want to perform a reverse process over $T$ iterative refinement steps, i.e. the goal is to go from a randomly sampled Gaussian noise back to the image by iterative denoising. To do so, we first need to approximate $\mathbf{y}_0$ by utilizing Equation (8) to obtain:

$$\hat{\mathbf{y}}_0 = \frac{1}{\sqrt{\gamma_t}}\left(\mathbf{y}_t - \sqrt{1-\gamma_t}f(\mathbf{x}, \mathbf{y}_t, \gamma_t)\right). \tag{10}$$

Now we can substitute $\hat{\mathbf{y}}_0$ into $p(\mathbf{y}_{t-1}|\mathbf{y}_0, \mathbf{y}_t)$ from Equation (7) in order to parameterize the mean of $p_\theta(\mathbf{y}_{t-1}|\mathbf{y}_t, x)$ as:

$$\mu_\theta(\mathbf{x}, \mathbf{y}_t, \gamma_t) = \frac{1}{\sqrt{\alpha_t}}\left(\mathbf{y}_t - \frac{1-\alpha_t}{\sqrt{1-\gamma_t}} f_\theta(\mathbf{x}, \mathbf{y}_t, \gamma_t)\right). \tag{11}$$

By setting the variance of $p_\theta(\mathbf{y}_{t-1}|\mathbf{y}_t, \mathbf{x})$ to $(1-\alpha_t)$, each step of the iterative reverse process can be computed as:

$$\mathbf{y}_{t-1} \longleftarrow \frac{1}{\sqrt{\alpha_t}}\left(\mathbf{y} - \frac{1-\alpha_t}{\sqrt{1-\gamma_t}} f_\theta(\mathbf{x}, \mathbf{y}_t, \gamma_t)\right) + \sqrt{1-\alpha_t}\boldsymbol{\epsilon}_t. \tag{12}$$

Pseudocode describing the inference procedure is given in Algorithm 4.

---

**Algorithm 4** Inference - Reverse (Denoising) Process

---

1: $\mathbf{y}_T \sim \mathcal{N}(\mathbf{0}, \mathbf{I})$
2: **for** $t = T, \ldots, 1$ **do**
3:     $\mathbf{z} \sim \mathcal{N}(\mathbf{0}, \mathbf{I})$ if $t > 1$, else $\mathbf{z} = \mathbf{0}$
4:     $\mathbf{y}_{t-1} = \frac{1}{\sqrt{\alpha_t}}\left(\mathbf{y}_t - \frac{1-\alpha_t}{\sqrt{1-\gamma_t}} f_\theta(\mathbf{x}, \mathbf{y}_t, \gamma_t)\right) + \sqrt{1-\alpha_t}\mathbf{z}$
5: **end for**
6: **return** $\mathbf{y}_0$

---

## B  Implementation Details

### B.1  Architecture and Hyperparameters

We base our U-Net architecture on Saharia et al. (2022) with modifications listed in Section 3.1. Following Karras et al. (2022), a linear noise scheduling is applied for the diffusion process spanning (1e-6, 0.01) over 2000 timesteps, both for training and inference.

We train the model using $L_2$ loss computed between the loss prediction and true noise. Furthermore, a learning rate scheduler is used in combination with Adam optimizer. The learning rate starts at 5e-5 with a 10k steps as a warm-up following which it peaks at 1e-4. The model is conditioned on one to six input views and trained for 710k steps using a batch size of 112 and 4×V100 GPUs. The total training time using this setup amounts to approximately 6.5 days.

Listing 1 shows PyTorch pseudocode for aggregating the view contributions at each diffusion step, given an arbitrary, unordered and pose-free collection of input views.

At inference time, we run the model for 2000 timesteps which takes around 2 minutes and does not depend on the amount of views used for conditioning (as long as they fit in the memory) since all the streams are treated as a batch. Using a single 32GB V100, we are able to process a batch size of 28 with up to six conditional input views, meaning that our model is able to process up to 168 64×64 images at a time.

### B.2  Evaluation Details

In order to ensure consistency of our evaluation process given the stochastic generative nature of our model, and while maintaining a reasonably low computational footprint, we repeat the evaluation procedure several times using the same model. For single-view setting, we evaluate the same model three times over the whole test dataset, by randomly picking an input view and an arbitrary target for each object. Table 2 reports the mean metrics of this procedure. We omit reporting standard deviations directly in Table 2 as they are orders of magnitudes lower than the metrics themselves, namely ±3.18e-2 for PSNR, ±4.78e-4 for SSIM and ±1.41e-4 for LPIPS, respectively. We perform a single run where the model receives up to six views, as it is significantly more expensive than the single-view setting and its results cannot be directly compared to prior methods.

It is important to note that our evaluation procedure differs slightly from evaluation procedures of Sajjadi et al. (2022b); Lin et al. (2023) where for a randomly picked input view, all other 23 views are generated. We

avoid performing such procedure as we observe stable results are already obtained from the separate runs, but also to maintain a low computational footprint as our sampling procedure can take a significant amount of time to be computed over the whole dataset.

**Listing 1** PyTorch Pseudocode for Contribution Aggregation

```
1
2  # create delimiter indices for unstacking the U-Net outputs
3  # view_count of shape (B, 1) - number of input views for each sample in the batch
4  view_delimiters = torch.cumsum(view_count, 0).tolist()
5  view_delimiters.insert(0, 0)
6
7  noise_level = extract(self.gammas, t, x_shape=(1, 1)).to(y_t.device)
8
9  # prepare shapes of conditioning, noise, angles and levels;
10 # from (B, 23, C, H, W) ->  ((V_1 + ... + V_B), C, H, W); where v_n is the input view
       count for each sample
11 x_stacked = torch.concatenate(
12     [x_i[i, :idx] for i, idx in enumerate(view_count)],
13     dim=0,
14 ).to(x_i.device)
15
16 y_t_stacked = torch.repeat_interleave(y_t, view_count, dim=0)
17 noise_level_stacked = torch.repeat_interleave(noise_level, view_count, dim=0)
18 angle_stacked = torch.repeat_interleave(angle, view_count, dim=0)
19
20 unet_output = self.unet(
21     torch.cat([x_stacked, y_t_stacked], dim=1),
22     angle_stacked,
23     noise_level_stacked,
24 )
25 noise_all, logits = unet_output[:, :3, ...], unet_output[:, 3:, ...]
26
27 # weights and noise padded; shape (B, max(V_1, ... , V_B), C, H, W)
28 logits_padded = torch.nn.utils.rnn.pad_sequence(
29     [
30         logits[idx1:idx2]
31         for idx1, idx2 in zip(view_delimiters[:-1], view_delimiters[1:])
32     ],
33     batch_first=True,
34     padding_value=float("-inf"), # -inf padding becomes 0 after softmax
35 )
36 weights_softmax = F.softmax(logits_padded, dim=1)
37 noise_padded = torch.nn.utils.rnn.pad_sequence(
38     [
39         noise_all[idx1:idx2]
40         for idx1, idx2 in zip(view_delimiters[:-1], view_delimiters[1:])
41     ],
42     batch_first=True,
43 )
44 noise_weighted = noise_padded * weights_softmax
45
46 noise = noise_weighted.sum(dim=1)
```

## C   A Probabilistic Interpretation of ViewFusion

We hereby elucidate the design choices of the *ViewFusion* approach by means of a probabilistic interpretation. In the following, we will assume a specific expression for the transition probability of the reverse diffusion process, where we will recover a weighted composition of per-view noise gradients.

We use the following notation:

1. $\mathbf{y}_{t-1}$ and $\mathbf{y}_t$, the images produced by the diffusion process at times $t-1$ and $t$, respectively;

2. $t$ being the time variable;

3. $\{\mathbf{x}_i\}_{i=1}^N$ being the set of $N$ conditioning images, each detailing a different pose of the object of interest;

4. $\Delta\psi$ being the conditioning target angle.

For brevity, we contain all the conditional information in a tuple $\mathbf{c}_i^t := (\mathbf{x}_i, \mathbf{y}_t, \Delta\psi, t)$.

We want to model the transition probability of the reverse diffusion process

$$p(\mathbf{y}_{t-1}|\mathbf{c}^t).$$

Since $\{\mathbf{x}_i\}_{i=1}^N$ is a set, the expression of the probability $p(\mathbf{y}_{t-1}|\mathbf{c}_i^t)$ should not depend on the order of the conditioning images, nor on their number. We can consider a single function $E$ applied separately to each of the $N$ views, and have these terms contribute to the final probability via summation. In a formula,

$$p(\mathbf{y}_{t-1}|\mathbf{c}^t) \propto \sum_{i=1}^N \exp(-E(\mathbf{y}_{t-1}, \mathbf{c}_i^t)).$$

With this prescription, in accordance to Zaheer et al. (2017), the probability is now a function that does not depend on the order of the $N$ views, and can be applied to any number of views (in their notation, we have $\phi = \exp(-E)$ and $\rho = \text{Id}$). In the machine learning context, this formulation is sometimes referred to as a *Mixture of Experts (MoE)*, where the experts here correspond to the different input views, and the output is the probability distribution of the reverse diffusion step. This is also known, in the context of physics, as a *Boltzmann distribution*. The analogy implies that the $N$ views act as *states* of the system, each of them with an associated *energy*, and the probability of each state is proportional to the exponential of the negative of this energy. These states then combine their influence on the reverse diffusion process paths via their summation.

We are now interested in the *score*,

$$\nabla_{\mathbf{y}} \log p(\mathbf{y}_{t-1}|\mathbf{c}^t).$$

as this is the quantity which the model is trained to predict, following Song et al. (2020b). We perform the computations directly:

$$\nabla_{\mathbf{y}} \log p(\mathbf{y}_{t-1}|\mathbf{c}^t) = \nabla_{\mathbf{y}} \log \sum_{i=1}^N \exp(-E(\mathbf{y}_{t-1}, \mathbf{c}_i^t)) \tag{13}$$

$$= \frac{\nabla_{\mathbf{y}} \sum_{i=1}^N \exp(-E(\mathbf{y}_{t-1}, \mathbf{c}_i^t))}{\sum_{j=1}^N \exp(-E(\mathbf{y}_{t-1}, \mathbf{c}_i^t))} \tag{14}$$

$$= \sum_{i=1}^N \frac{\nabla_{\mathbf{y}} \exp(-E(\mathbf{y}_{t-1}, \mathbf{c}_i^t))}{\sum_{j=1}^N \exp(-E(\mathbf{y}_{t-1}, \mathbf{c}_i^t))} \tag{15}$$

$$= \sum_{i=1}^N \frac{\exp(-E(\mathbf{y}_{t-1}, \mathbf{c}_i^t))}{\sum_{j=1}^N \exp(-E(\mathbf{y}_{t-1}, \mathbf{c}_i^t))} \nabla_{\mathbf{y}}(-E(\mathbf{y}_{t-1}, \mathbf{c}_i^t)), \tag{16}$$

where we recognize the softmax function. For ease of writing and reading, we call

$$\mathbf{w}_i = \frac{\exp(-E(\mathbf{y}_{t-1}, \mathbf{c}_i^t))}{\sum_{j=1}^N \exp(-E(\mathbf{y}_{t-1}, \mathbf{c}_i^t))}. \tag{17}$$

Then

$$\nabla_{\mathbf{y}} \log p(\mathbf{y}_{t-1}|\mathbf{c}^t) = \sum_{i=1}^N \mathbf{w}_i \nabla_{\mathbf{y}}(-E(\mathbf{y}_{t-1}, \mathbf{c}_i^t)), \tag{18}$$

and so we recover the weighting of the contributions of each conditioning image.

While, in principle, knowledge of $E$ is enough to completely characterize $p$ (as each of the $\mathbf{w}_i$ is expressible as a function of all the $N$ energies), in this work we predict separately the weights $\mathbf{w}_i$ and the gradients of the energies (also called scores).

# D    Additional Results

## D.1    Relative Canonical Pose

Defining the canonical view as the first, front facing view of the object at 0° as given by the dataset assumes similar objects to have the same canonical pose, resulting in a close relation between the semantics and the camera pose. Instead of assuming the canonical view to be defined by the dataset, here we test our model's performance when canonical pose is defined relative to the first input view. We assume the first of the input views, $\mathbf{V}_1$, is the reference view and define $\Delta\psi$ as angular disparity between the target view and the first input view $\mathbf{V}_1$. The model is trained from scratch under this premise.

We observe that our approach handles this case successfully as well, producing sensible results (see Figure 8). However, the metrics computed on a validation subset of NMR dataset are lower: 31.17 (canonical) vs. 26.54 (relative) in PSNR and 0.95 (canonical) vs. 0.89 (relative) in SSIM. We don't compute LPIPS at validation time. The comparably lower performance can be attributed to this setup posing a considerably more difficult task.

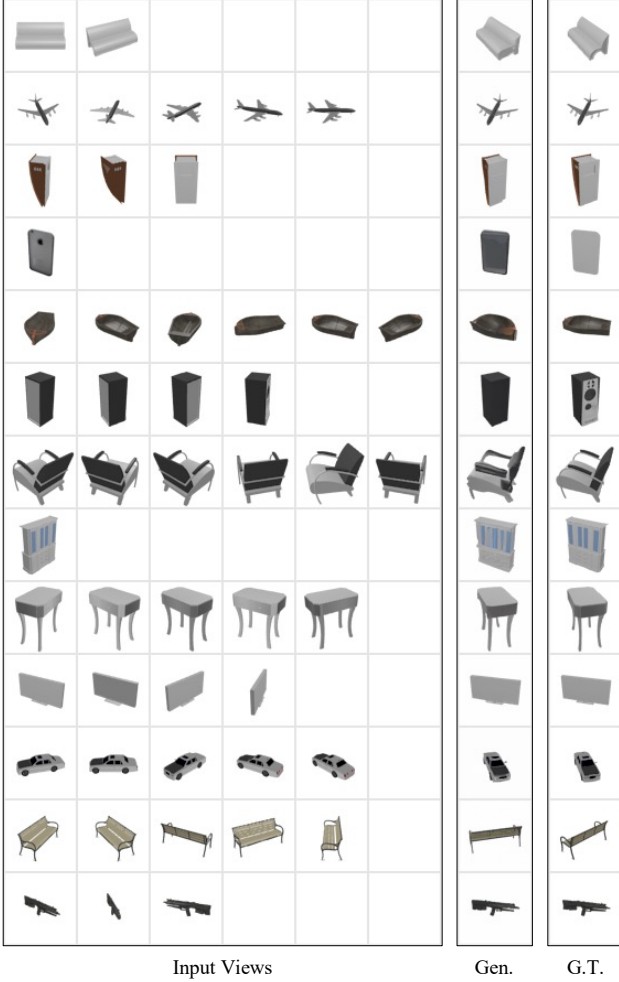

Input Views                                    Gen.          G.T.

Figure 8: **Relative Canonical Pose.** Instead of defining $\Delta\psi$ as the front facing view of the object as defined in the dataset we assume the first of the input views, $\mathbf{V}_1$, is the reference view and define $\Delta\psi$ as angular disparity between the target view and the first input view $\mathbf{V}_1$.

## D.2 Generalization to Unseen Objects

We test our method's generalization to previously unseen classes by training the model from scratch and omitting the car class entirely from the training set. Then, we run autoregressive inference using an image of a car and show that the model is able to rotate a car up to 180° fairly reasonably, but beyond that it collapses (starting from rear facing view) as shown in Figure 9. While the results are far from perfect, it is interesting to observe that the model still captures relatively meaningful information from images of unknown objects, especially given that the entire car class was not present in the training set.

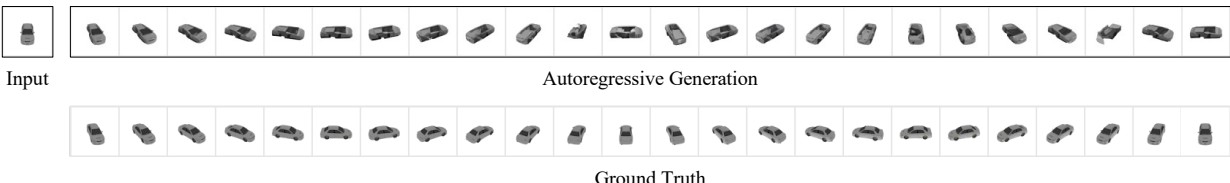

Input                                                                 Autoregressive Generation

Ground Truth

Figure 9: **Autoregressive out of Distribution Generation.** We first omit the car class entirely from the training set. We then once again prime the model with a single input view of a car, and incrementally rotate the target viewing direction to produce novel views. During the autoregressive generation, each consecutively generated view is added to the flexible conditioning for producing the next view.

## D.3 Weighting Scheme Ablation

Here we justify the design decision to use a weighting mechanism by performing a set of ablation experiments.

In the first experiment, we train the model from scratch using simple averaging to combine the noise contributions and completely omit the weighting scheme. We observe that the averaged model is consistently trending lower than the weighted model on the validation metrics as shown in Figure 10.

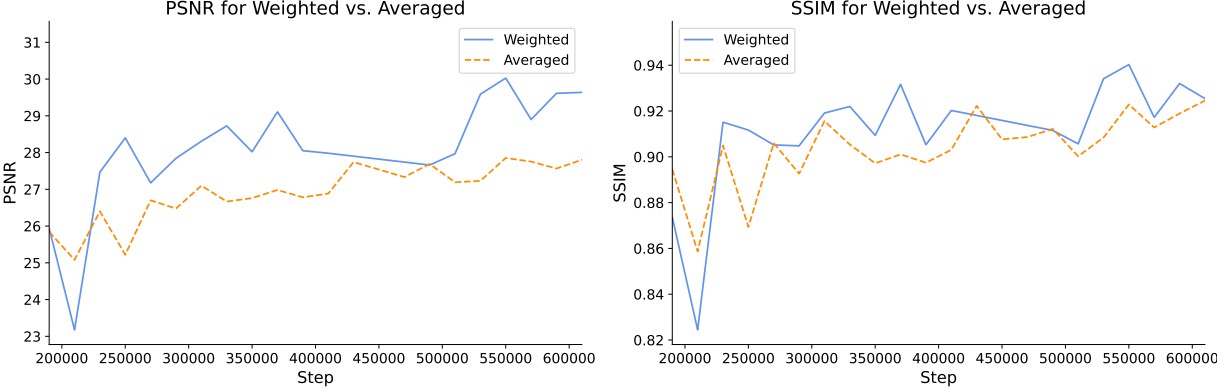

Figure 10: **Weighting vs. Simple Averaging.** We train the model from scratch using simple averaging to combine the noise contributions and completely omit the weighting scheme. We observe that the averaged model is consistently trending lower than the weighted model on the validation metrics, with a more significant difference in PSNR.

In the second experiment, we simply disable the weighting scheme of an already trained model and just averaged the noise contributions. We note that our model's capability to operate well in a single view setting, which is equivalent to the weighting scheme being off, also ensures that each of the individual streams in a multi view setting produce sensible result on their own. Then, the weighting scheme helps with picking out the best information from each of the streams meaning that simple averaging should still produce sensible

results. Replacing the weighting with averaging in an already trained model also results in a performance drop on validation set both in PSNR - 27.38 (weighted) vs. 25.88 (averaged) as well as in SSIM - 0.90 (weighted) vs. 0.88 (averaged).

