# OpenReview forum: "ViewFusion: Learning Composable Diffusion Models for Novel View Synthesis"
_TMLR — Accepted by TMLR_

### Review · Reviewer_yyPb · 2025-02-18

**Summary Of Contributions:**

This paper presents a new multi-view diffusion method to synthesize novel views from pose-free and length-varied image series. The main contribution is to predict the noise as well as a weight according to each input image and the target pose, and use Softmax to merge these noises for the current timestep. Experiments on NMR datasets show that it gets competitive results compared to some generative NeRF-based methods.

**Audience:**

Yes

**Claims And Evidence:**

No

**Requested Changes:**

Major:

1. Add discussions and comparisons with the advanced relevant methods.

2. If available, add more evaluations with different datasets and resolutions to verify the scalability.

3. Clarify the handling of camera poses.

**Strengths And Weaknesses:**

Strengths:

1. It allows the input of unordered and pose-free collections of images with variable length, showing the advantage compared to the NeRF-based baselines.
2. The method is simple and direct. It may be easy to be applied in other works.

Weaknesses:

1. The paper lacks discussions with some relevant works like Diffusion-based [1] and NeRF-based [2]. Especially,[1] proposes very similar idea to this paper, which also achieves novel view synthesis with unordered and pose-free collections of images with variable length by fusing the input views. It should be revisited whether the idea and declared contributions are still interesting in the field.

2. The evaluation is somewhat weak. First, the method is just evaluated on NMR with a resolution of 64x64, of which the data diversity is far from the prevailing datasets like Objaverse and Google Scanned Object. And the resolution is also much lower than the prevailing setting of 256x256 and 512x512. It is doubtful whether the proposal is scalable to larger data scenarios. Second, the baselines are somewhat outdated. The newest baseline in the paper is ViT for NeRF, which is released on arxiv in 2022.07. It's expected to verify if the performance is competitive with the advanced methods like mentioned in 1.

3. I'm curious about the solution for the different camera coordinates between scenes. In 3.1, it said, "∆ψ is the angular disparity between the target view and the canonical pose of the object, i.e. the front facing view of the object as defined in the dataset". That's to say, the current solution seems to hypothesize the canonical pose of similar objects to be the same, like all the cars should have a same canonical pose like facing the front of the car, while the canonical pose of a table may be from another specific angle. This defines a tight relation between the semantics and the camera pose. However, as the method is pose-free, it should be learned by the model. This is weird, which heavily restricts that all the scenes in the training dataset have a well-calibrated canonical pose. And as the pose is ambiguous, the model seems not to be very robust for out-of-domain semantics, which makes it hard even to specify a target view.

[1] Tang, Shitao, et al. "Mvdiffusion++: A dense high-resolution multi-view diffusion model for single or sparse-view 3d object reconstruction." European Conference on Computer Vision. Cham: Springer Nature Switzerland, 2024.

[2] Jiang, Hanwen, et al. "LEAP: Liberate Sparse-View 3D Modeling from Camera Poses." The Twelfth International Conference on Learning Representations.

---

> ### Author Response · Authors · 2025-03-07
>
> Dear reviewer,
>
> We would like to thank you for the review and valuable comments as well as acknowledging the simplicity and directness of our approach.
> We address the requested changes below:
>
> **1. Add discussions and comparisons with the advanced relevant methods.**
>
> Thank you for pointing out the relevant works that we have now added to the "Related Work" section.
> Despite the potential similarities, we believe that the idea and our contributions are interesting for the field for a number of reasons.
>
> After carefully going through contributions of MVDiffusion++, the advantages and differences that we see our work provides are:
>
> 1. MVDiffusion++ approach operates on a fixed, predefined grid. This means that if we want to increase the fine-grain control over the angles the model is capable to produce, the grid needs to become increasingly dense.
> 2. Relatedly, to our understanding, producing a specific view at inference time using MVDiffusion++, requires generating a full grid of 32 views and then pick the target view manually. Contrary to that, our approach allows for direct control over the desired output view through specifying a target angle. Consequently, this can result in a significantly lower computational overhead.
> 3. They state that "The number of condition images is up to a pre-determined number, which is 10 in our experiments but can easily change." - This means that, unlike in our approach, once the model is trained it cannot adaptively make use (extrapolate) of more input views than the maximum provided at training time.
> 4. MVDiffusion++ relies on CLIP embeddings, and our model can be trained in a straightforward manner from scratch.
>
> Likewise, when comparing to LEAP's approach, they are the following:
>
> 1. Based on their results it seems like LEAP performs well with five input views (k=5), but relatively poorly when given only a single view (k=1, shown in Appendix C in LEAP paper) whereas our model handles this scenario really well due to generative capabilities of diffusion.
> 2. Relatedly, LEAP uses ViT (DINOv2 initialized) embeddings which are deterministic, reducing the method's generative capabilities and performance on the occluded setting is not discussed in their work.
>
> => We have now included these discussion points in the "Related Work" section as well.
>
> **2. If available, add more evaluations with different datasets and resolutions to verify the scalability.**
>
> We agree with the reviewer that it would be a good idea to verify the method on a larger and higher-resolution dataset, as also pointed out in the "Limitations and Future Work" section.
>
> To this end, we did integrate our approach into Zero123 pipeline in order to apply it on Objaverse dataset. However, upon further testing, we realised that the computational resources needed to perform training and inference on such a large scale are unfortunately not available to us. Given that Zero123 does not offer the weighting scheme that we propose, thus requiring re-training (or fine-tuning) to address for that, which is very computationally expensive.
>
> We picked our baselines based on prior works that had evaluations on NMR available, as it was our dataset of choice - providing a good variety of classes and different objects available, while still offering acceptable computational footprint for our available resources.
>
> => We now further elaborate in "Limitations and Future Work" section mentioned large-scale datasets our method could be expanded on.
>
> **3. Clarify the handling of camera poses.**
>
> The reviewer is correct to state that the current solution hypothesises the canonical pose of similar objects to be the same, which is indeed a weakness of our approach. Currently the canonical pose was defined as a zero degree (first view) in the NMR dataset.
>
> => This is now better clarified in the "Method" section.
>
> We believe that this is a very good point and understand that using the canonical pose comes with some unwanted and vague assumptions.
> For this reason, we have now performed additional ablations, where instead of defining $\Delta \psi$ as "the front facing view of the object as defined in the dataset" we assume the first of the input views is the reference view and define $\Delta \psi$ as rotation with respect to it. We train the model from scratch under this premise.
>
> We observe that our approach handles this case successfully as well, producing sensible results (now included in Appendix D.1).
> However, the metrics computed on a validation subset of NMR dataset are lower: 31.17 (absolute) vs. 26.54 (relative) in PSNR and 0.95 (absolute) vs. 0.89 (relative) in SSIM. We don't compute LPIPS at validation time.
>
> We believe this is expected behavior as this setup poses a considerably more difficult task.
>
> => The results of the discussed setup are now included in the appendices and referred to in the "Method" section. We also discuss them in the "Limitations and Future Work" section.

---

### Review · Reviewer_duNQ · 2025-02-20

**Summary Of Contributions:**

This paper introduces a new model to generate a novel view image of an object from an arbitrary number of images of this object. The key idea is to adopt multiple Zero-123-like architectures to process each input view. Then, the predicted noises are averaged by a predicted weight for the denoising. In this scheme, the proposed method is able to process an arbitrary number of input views to generate novel view images.

**Audience:**

Yes

**Claims And Evidence:**

Yes

**Requested Changes:**

1. Improve the writing. The key idea of this paper is not very complex. I think we can say that the proposed method could be a combination of multiple Zero123 models to denoise a single-view image.
2. I would suggest directly fine-tuning Zero123 on Objaverse and demonstrating some generalization ability in real-world examples instead of just synthetic data.
3. If we use some input views, then it is possible to directly apply NeRF/3DGS. Some recent works like GaussianObject are able to reconstruct the object in just 4 views.
4. If we generate images in an autoregressive manner, how the errors accumulate in the whole process is unknown. It would be better to check the 3D reconstruction quality so that we can better see the 3D consistency of the proposed method.
5. Many related works about multiview diffusion are not discussed like MVDream, SyncDreamer. And some sparse-view reconstruction methods are not discussed either, like Instant3D.

**Strengths And Weaknesses:**

Strength:
1. The problem of processing an arbitrary number of input images for novel-view images is interesting.
2. The idea of combining the predicted noises from multiple zero123 models is simple but plausible.

Weakness:
1. Some claims are not accurate: the proposed method still requires pose information to define the pose difference $\Delta \psi$. However, the paper claims the method is able to handle pose-free images.
2. The results of experiments are not impressive enough. The results all are based on synthetic data while the original Zero123 shows the ability to rotate a 2D object even in an oil painting style.
3. The writing of the paper could be improved. The first section of the method describes lots of implementation details but what we need to understand is the high-level idea of the paper. The introduction is not a good overview of the proposed method.

---

> ### Author Response · Authors · 2025-03-07
>
> Dear reviewer,
>
> Thank you for the review and valuable comments, to which we answer below:
>
> **1. Improve the writing. The key idea of this paper is not very complex. I think we can say that the proposed method could be a combination of multiple Zero123 models to denoise a single-view image.**
>
> We have now improved the introduction to better reflect the high-level idea of the work:
>
> > We propose ViewFusion, a novel approach that tackles the mentioned drawbacks all at once through a series of problem-specific design choices. Our method employs a diffusion probabilistic framework. We simultaneously apply a diffusion denoising step to any number of input views of a scene, then combine the noise gradients obtained for each view with a pixel-weighting mask, inferred specifically for every view, to ensure that for each region of the target view only the most informative input views are taken into account. Our method can be understood as a combination of multiple single-view diffusion models for novel view synthesis, which produce weights used to aggregate the corresponding noise predictions of each of the single-view diffusion models during the denoising process.
>
> => We now clarify the high-level idea of the method in "Introduction" section.
>
> **2. I would suggest directly fine-tuning Zero123 on Objaverse and demonstrating some generalization ability in real-world examples instead of just synthetic data.**
>
> We agree with the reviewer's suggestion to integrate our approach into Zero123 to demonstrate large-scale generalization capabilities, as we also pointed this out as one of the drawbacks of our method in the "Limitations and Future Work" section.
>
> To this end, we integrated our approach into Zero123 pipeline in order to apply it on Objaverse dataset. However, upon further testing, we realised that the computational resources needed to perform training and inference on such a large scale are unfortunately not available to us. Given that Zero123 does not produce the per-pixel weighting scheme by design, thus requiring re-training (or fine-tuning) to address for that, which is very computationally expensive.
>
> NMR dataset provided a good variety of classes and different objects available, while still offering acceptable computational footprint for our available resources.
>
> => We now further elaborate in "Limitations and Future Work" section the mentioned large-scale datasets our method could be expanded on as well as integration possibilities to existing works.
>
> **3. If we use some input views, then it is possible to directly apply NeRF/3DGS. Some recent works like GaussianObject are able to reconstruct the object in just 4 views.**
>
> NeRF and 3DGS are indeed very powerful methods and have evolved to be able to produce very high quality reconstructions from very sparse views. However, they are still lacking when it comes to generative power and handling occluded areas in the scene. This is where we believe diffusion and other similar generative approaches shine.
>
> **4. If we generate images in an autoregressive manner, how the errors accumulate in the whole process is unknown. It would be better to check the 3D reconstruction quality so that we can better see the 3D consistency of the proposed method.**
>
> We are unable to perform this with our method as it is currently, since it operates purely in image space and doesn't produce full 3D reconstructions / point clouds of objects. The lack of incorporating 3D scene semantics is something we mention as a drawback, also citing it as a potential future work in "Limitations and Future Work" section.
>
> **5. Many related works about multiview diffusion are not discussed like MVDream, SyncDreamer. And some sparse-view reconstruction methods are not discussed either, like Instant3D.**
>
> Thank you for noting these works, we have now expanded the discussion in "Related Work" to include them. We also note that these works are on text-to-view synthesis (except SyncDreamer which also supports single-view setting), whereas our method focuses on view-to-view synthesis.
>
> => We now include MVDream, SyncDreamer, Instant3D works in "Related Work" section.

---

> ### Author Response · Authors · 2025-03-07
>
> **Some claims are not accurate: the proposed method still requires pose information to define the pose difference $\Delta\psi$. However, the paper claims the method is able to handle pose-free images.**
>
> The reviewer is correct to state that our method requires pose information for the target view.
> Indeed, like other pose-free methods, our method requires to somehow specify the target pose, such that the model is instructed of which view to produce.
> We specify the target pose as an angular difference relative to the canonical pose of the object, as defined by the viewing angle of 0° for that object according to the dataset NMR.
> Like for other pose-free methods, the "pose-free" qualifier refers to the fact that the input views are given without pose information or any ordering, and their number is not fixed.
> For example, SRT is a pose-free method, which nonetheless requires query rays coordinates as a means to specify the target pose.
>
> In response to another comment by Reviewer yyPb, we now also include additional results (Appendix D.1) where the target pose is defined as an angular difference w.r.t. the first input view. Our model handles this case successfully as well, while performing slightly worse on the validation set: 31.17 (absolute) vs. 26.54 (relative) in PSNR and 0.95 (absolute) vs. 0.89 (relative) in SSIM. We don't compute LPIPS at validation time. We believe this is expected behavior as this setup poses a considerably more difficult task.
>
> => We clarified in main text how the target pose is being specified relative to a canonical pose for the object.

---

### Review · Reviewer_ahEw · 2025-03-25

**Summary Of Contributions:**

This paper presents a generative method for novel view synthesis based on diffusion models. The base model is a standard diffusion u-net that predicts one output view given an input view and relative pose. However the method fuses information from several model instances during both training sampling, allowing varying numbers of input images from different viewpoints, and avoiding any order dependence. Specifically, a weighting is predicted in addition to the noise, and used to determine the influence of different input views at each output pixel. The method is evaluated on the NMR dataset of ShapeNet renderings, where it achieves better performance than three baseline approaches.

**Audience:**

Yes

**Broader Impact Concerns:**

There is a broader impact statement present; however this work does not raise any specific concerns.

**Claims And Evidence:**

No

**Requested Changes:**

Justify or weaken the not-fully-supported claims mentioned above under 'weaknesses'

Add qualitative results for baselines

If possible, include some ablation / variant experiments to more clearly justify the design decisions made in the proposed approach

**Strengths And Weaknesses:**

**Strengths**

- The underlying idea of combining the information from several instances of a single-image NVS diffusion model during sampling is elegant, and to the best of my knowledge novel in this application
- The specific approach of learning a noise-weighting at each pixel (determining how much each single-view u-net affects the output) is also interesting (and again novel in this setting)
- Quantitative results on ShapeNet renderings show the proposed approach exceeding three baselines (Light Field Nets, pixelNeRF, SRT, ViT-for-NeRF) on the standard reconstruction metrics (PSNR, SSIM, LPIPS)
- Qualitative results appear high quality, with sharp and plausible synthesised novel views that appear consistent with the input image
- There are also results demonstrating that the model behaves generatively, i.e. can sample diverse (but still plausible) completions for ambiguous, self-occluded regions
- There is a fairly detailed discussion of limitations
- Overall the text is clear and readable; the paper is well structured; the figures are appropriate

**Weaknesses**

- Experiments are only conducted on ShapeNet (specifically, the renderings from NMR); this is a rather small and simplistic dataset by modern standards. It would have been good to see experiments on a more challenging dataset – either more diverse synthetic data (Objaverse or similar), or more complex single-class data (CO3D, MVImageNet or similar)
- There are no ablation experiments, or experiments using variations of the proposed model, to justify its design decisions. In particular, the novel fusion approach is interesting, but it'd be good to see quantitatively the benefit of using the predicted weights (rather than equal weighting), and also to the see performance of an identical architecture (on identical data) but using a more 'traditional' fusion approach (e.g. concatenating a fixed number of input views, or performing cross-attention)
- There is no qualitative comparison against the baselines, making it difficult to intuitively understand how the proposed method performs in comparison
- A key claim (and motivation for the architecture) is the support for varying numbers of views; however this is only backed up with qualitative evidence in the evaluation. It would be good to see quantitative results (maybe a graph?) on how view-synthesis performance varies with the number of views, with both fewer and more views than used during training. Presumably there is some 'overfitting' to the view count seen during training, even if the model adapts relatively smoothly to other counts?
- The claim in sec. 5 that the method supports generation of explicit 3D representations (rather than just independent images that might be 3D-inconsistent across output views) is justified only by referencing qualitative results on autoregressive image generation. To justify this claimed application properly, an attempt should be made to actually perform a reconstruction on these images (e.g. using NeRF) and render this from interpolated viewpoints
- The same is true of the claim that the method can be used for data augmentation – this may be true, but there is no evidence – and synthetic shapenet renderings can be created almost for free, so on the specific domain where the method is demonstrated, this does not seem useful
- The baselines are all non-generative – being deterministic, they will necessarily tend to make blurry predictions in uncertain regions, and thus the comparison is not entirely fair. It'd be good to include some existing generative approach as an additional baseline
- Although the approach is generative, there is no quantitative measurement of diversity – there are three qualitative examples in fig 4, but nothing more

---

> ### Author Response · Authors · 2025-04-10
>
> Dear reviewer,
>
> We would like to thank you for the exhaustive review and the insightful comments.
> We address the questions and requests below:
>
> **Experiments are only conducted on ShapeNet (specifically, the renderings from NMR); this is a rather small and simplistic dataset by modern standards. It would have been good to see experiments on a more challenging dataset – either more diverse synthetic data (Objaverse or similar), or more complex single-class data (CO3D, MVImageNet or similar)**
>
> We agree with the reviewer that experimenting on a more diverse and challenging dataset would provide additional validation of our method. We did attempt to integrate our approach into the Zero123 pipeline to apply it on the Objaverse dataset. However, the computational resources required for training and inference on such a large scale were not available to us.
>
> We chose the NMR dataset for its variety of classes and objects, while still offering an acceptable computational footprint for our available resources. We acknowledge that expanding our evaluations to more complex datasets like CO3D, Objaverse or MVImageNet would be beneficial and have noted this in the "Limitations and Future Work" section.
>
> => "Limitations and Future Work" section now further elaborates the mentioned large-scale datasets our method could be expanded on as well as integration possibilities to existing works.
>
> **There are no ablation experiments, or experiments using variations of the proposed model, to justify its design decisions. In particular, the novel fusion approach is interesting, but it'd be good to see quantitatively the benefit of using the predicted weights (rather than equal weighting), and also to the see performance of an identical architecture (on identical data) but using a more 'traditional' fusion approach (e.g. concatenating a fixed number of input views, or performing cross-attention)**
>
> We thank the reviewer for this insightful suggestion as these experiments are indeed an interesting direction to explore. To this end, we have now performed two additional ablation experiments where we disable our weighting scheme and just average the noise contributions.
>
> In the first experiment, we trained the model from scratch with simple averaging, completely omitting the weighting scheme and its training. We observe that the averaged model is consistently trending lower than the weighted model on validation metrics.
> In the second experiment, we simply disabled the weighting scheme of an already trained model and just averaged the noise contributions. In this scenario we also observe a performance decrease when the weighting mechanism is off in PSNR - 27.38 (weighted) vs. 25.88 (averaged) as well as SSIM - 0.90 (weighted) vs. 0.88 (averaged). The results of the ablation are now available in the Appendix D.3.
>
> We note that during the development of this method, we experimented with quite a few fusion approaches, like concatenation and fusion of input view latents within the U-Net, before deciding on the approach proposed here as it offered the best performance and interpretability while simultaneously being very straightforward and easy to apply to existing methods. We did not explore these attempts sufficiently so that we could add them to the paper as they elicited poor performance from the very beginning.
>
> => We now include two additional ablations to justify the weighting mechanism that we propose.
>
> **There is no qualitative comparison against the baselines, making it difficult to intuitively understand how the proposed method performs in comparison**
>
> We agree that having more qualitative comparisons would be helpful. However, this is technically overly challenging as it would require us to re-run the baselines locally, supposing that their code is available and works out of the box in the first place, which is not always the case.

---

> ### Author Response · Authors · 2025-04-10
>
> **A key claim (and motivation for the architecture) is the support for varying numbers of views; however this is only backed up with qualitative evidence in the evaluation. It would be good to see quantitative results (maybe a graph?) on how view-synthesis performance varies with the number of views, with both fewer and more views than used during training. Presumably there is some 'overfitting' to the view count seen during training, even if the model adapts relatively smoothly to other counts?**
>
> We have now performed additional experiments where we gradually go from one to 9 views (beyond that the scene is over-determined, as chances of getting a very close view to the target become very high). The results show that the increased view count results in consistent performance increase both in PSNR and SSIM. The graph is now available in the "Results" section under "4.2 Evaluation Procedure".
>
> Note that our claims are also supported with quantitative results in Table 2 where we compare the model's performance with only a single view to performance when given a variable amount of views.
>
> => We now include quantitative analysis showing that the increasing the number of input views also improves the performance of the model.
>
> **The claim in sec. 5 that the method supports generation of explicit 3D representations (rather than just independent images that might be 3D-inconsistent across output views) is justified only by referencing qualitative results on autoregressive image generation. To justify this claimed application properly, an attempt should be made to actually perform a reconstruction on these images (e.g. using NeRF) and render this from interpolated viewpoints**
>
> We agree with the reviewer, our method is currently not able to generate explicit 3D representations. We have now updated the claims in section 5 to account for potential 3D inconsistencies as well as the method's capability to only produce independent images, rather than full 3D representation.
>
> => We have now corrected the claims to better reflect the capabilities of our method and outline that it only operates in the image domain, rather than on explicit 3D representations.
>
> **The same is true of the claim that the method can be used for data augmentation – this may be true, but there is no evidence – and synthetic shapenet renderings can be created almost for free, so on the specific domain where the method is demonstrated, this does not seem useful**
>
> => We have now modified the writing to better reflect the circumstances in which this would be applicable, e.g. if the method was upscaled to be capable of generating more realistic objects.
>
> **The baselines are all non-generative – being deterministic, they will necessarily tend to make blurry predictions in uncertain regions, and thus the comparison is not entirely fair. It'd be good to include some existing generative approach as an additional baseline**
> **Although the approach is generative, there is no quantitative measurement of diversity – there are three qualitative examples in fig 4, but nothing more**
>
> We have now added all the generative methods available to the best of our knowledge in "Related Work" section: Diffrf lacks the ability to generalize across different classes, Multidiff is pose-conditional, NVS with diffusion models, Renderdiffusion, Zero-1-to-3, Zero123++, and Cascade-zero123 cannot take multiple input views, Denoising diffusion via image-based rendering lacks the ability to extrapolate at test time beyond number of input views available at training time, and lastly Mvdiffusion++ operates on a fixed grid
> with pre-determined number of input views resulting in reduced flexibility.
> We note that none of these methods evaluate their performance on NMR, making it impossible to perform a fair comparison. In the absence of comparison, we decided not to investigate the diversity of the output.
>
> => Expanded Related Work section to include all relevant generative works.

---

### Decision · Action_Editor_iPpp · 2025-05-02

**Recommendation:** Accept as is

**Comment:**

Two of the three reviewers recommended acceptance, describing the approach as elegant, interesting, and simple. The reviewers also indicate some skepticism regarding ShapeNet-only experiments and missing competition with industry-style approaches that are more resource-intensive, but they also note that the paper is very direct about its limitations -- even stating some within the abstract, which is rare and useful. The AE sides with the majority and recommends acceptance without further review necessary, but the authors are encouraged to integrate the work presented in the rebuttal, as the reviewers found these additions helpful.

**Audience:**

Yes

**Claims And Evidence:**

Yes

---

> ### Author Response · Authors · 2025-05-20
>
> Thank you for the positive decision!
>
> We would like to once again thank the reviewers for thorough and insightful reviews provided, as well as the AE for making the whole review process seamless.
> We have now also fully integrated all the work presented in the rebuttal.
>
> Camera ready version has now been posted!